# Children’s Intake of Food from Non-Fast-Food Outlets and Child-Specific Menus: A Survey of Parents

**DOI:** 10.3390/children6110123

**Published:** 2019-11-01

**Authors:** Li Kheng Chai, Sze Lin Yoong, Tamara Bucher, Clare E Collins, Vanessa A Shrewsbury

**Affiliations:** 1Priority Research Centre for Physical Activity and Nutrition, School of Health Sciences, Faculty of Health and Medicine, University of Newcastle, Callaghan, New South Wales 2308, Australia; likheng.chai@uon.edu.au (L.K.C.); tamara.bucher@newcastle.edu.au (T.B.); clare.collins@newcastle.edu.au (C.E.C.); 2Hunter New England Population Health, Wallsend, New South Wales 2287, Australia; Serene.Yoong@health.nsw.gov.au; 3School of Environmental and Life Science, Faculty of Science, University of Newcastle, Ourimbah, New South Wales 2258, Australia

**Keywords:** childhood obesity, food environment, child menu, preschool, non-fast-food outlets

## Abstract

Eating out-of-home is associated with higher energy intakes in children. The continued high prevalence of childhood obesity requires a greater understanding of child menu options and eating out frequency to inform appropriate regulatory initiatives. The majority of studies to date have focused on menus from fast-food outlets with few focused on non-fast-food outlets. This study aimed to describe parents’ reports of their child(ren)’s (aged up to 6 years) frequency of consuming foods at non-fast-food outlets, observations of child menus at these outlets, and their purchasing behaviours and future preferences regarding these menus; and if their responses were influenced by sociodemographic characteristics. Ninety-five parents completed a 15-item cross-sectional survey. Overall, children from 54% of families consumed food from non-fast-food outlets at least monthly. Of the 87 parents who reported that their child eats at a non-fast-food restaurant, 71 had children who ordered from child menus every time (7%, *n* = 5), often (29%, *n* = 22), sometimes (42%, *n* = 32) or rarely (16%, *n* = 12), with a further 7% (*n* = 5) never ordering from these menus. All parents indicated that they would like to see a higher proportion of healthy child menu items than is currently offered. Parents’ responses were not influenced by sociodemographic characteristics. Parents’ views support implementation of initiatives to increase availability of healthy options on child menus at non-fast-food outlets.

## 1. Introduction

Population data from Australia, United States (US), and United Kingdom (UK) indicate that the majority of foods consumed are still prepared in the home setting although the consumption of food prepared outside the home has increased in recent decades [1,2,3,4]. Consuming food prepared outside of home and school settings (i.e., eating out) is associated with a greater intake of non-core foods [5] and contributes substantially to energy intake in children [6]. Data from the UK National Diet and Nutrition Survey Rolling Program (2008 to 2014) specifically identified that in children, 69% to 79% of eating occasions occurred in the home and 7% to 17% occurred in school. A further 5% to 7% of eating occasions occurred in leisure places, food outlets, and “on the go” combined. It was identified that foods consumed in these settings contribute a greater proportion of daily total energy intake from energy-dense, nutrient-poor, non-core foods as compared with nutrient-dense, core foods, with the opposite found in home or school settings [5]. 

Although studies have been conducted on the association between eating out-of-home and anthropometric changes or obesity risk, most have focused on fast-food outlets with few studies investigating this association in non-fast-food outlets [7,8]. Furthermore, some studies have identified that meals from full-service restaurants contain higher energy, fat, and salt than fast food restaurants [9,10,11,12]. These findings regarding the nutritional value of full-service restaurant foods are of particular interest because we have observed in New South Wales (NSW), Australia, that child menus at full-service (i.e., non-fast-food) restaurants contain only small quantities of nutrient-dense core foods, including vegetables, wholegrains, lean protein, dairy, and fruit, similar to findings in South Australian outlets [13]. 

The continued high prevalence of childhood obesity [14] requires a greater understanding of child menu options and eating out frequency to inform appropriate regulatory initiatives. To build a rigorous evidence base and inform effective interventions for improving the dietary quality of child menus at non-fast-food outlets, consumer engagement is key to developing a better understanding of the eating out-of-home habits and viewpoints of contemporary families. In Australia, such data are limited in the peer-reviewed literature. Several studies from the US have explored the perspectives of parents [15], children [15], and restaurant staff [15,16] on children’s meals in quick- or full-service restaurants [15]. The results indicated that almost two-thirds of children ordered from child menus, with 8% ordering healthier kids’ meals, despite healthier options being available in all participating food outlets. Interviews with restaurant staff highlighted the complex drivers of menu changes including profitability, customer demand, regulation, and corporate social responsibility [15,16].

To expand the evidence base, specifically regarding consumer views on child menus in the non-fast-food sector, the aim of this study was to describe parents’ reports of their young child(ren)’s (up to six years) frequency of consuming foods at non-fast-food outlets, their observations of child menus at these outlets and their purchasing behaviours and future preferences regarding these menus; and whether their responses varied by sociodemographic characteristics.

## 2. Materials and Methods

### 2.1. Ethics

This cross-sectional study was nested as part of follow-up data collection in a randomized controlled trial (RCT) for a web-based menu planning intervention to improve child dietary intake in childcare settings across NSW [17]. Ethical approval was provided by the Hunter New England Human Research Ethics Committee (approval No: 16/02/17/4.05) and the University of Newcastle Human Research Ethics Committee (approval H-2016-0111). The RCT is prospectively registered with the Australian New Zealand Clinical Trials Registry (ACTRN12616000974404) [17].

### 2.2. Procedures

Parents were recruited as part of an existing RCT [17]. Trial and data collection procedures are reported in detail in a published protocol [17]. Briefly, parents were recruited from consenting childcare centers within NSW by trained research assistants over one to two days during drop off time. Prior to the site visit, childcare services were asked to distribute information about the study and consent forms to parents via the usual communication means with parents, such as the child pigeonholes and electronically. As part of the study, parents were asked to provide informed consent to complete an online or telephone survey. Consenting parents were sent a link to complete an online survey. Up to two email reminders were sent prior to being provided the option of completing the survey over the telephone. Parents could choose to stop the survey at any point. No incentive was provided to participants in the study.

### 2.3. Measures

This study reports findings from 15 questions embedded within the larger online or telephone survey. Eight questions (Table 1) addressed the primary aim including six with categorical response options (Question 1 to 6) and two with continuous scales (Questions 7 and 8). Seven sociodemographic questions addressed child age and sex, parent sex, parent education, household annual income, living arrangements, and number of children in the family. A subsample of parents were randomly selected to receive these questions and were asked to respond with regard to their index child attending the childcare service. 

### 2.4. Statistical Analysis

All data manipulation and statistical analyses was undertaken using STATA version 12 (Stata Corp LP, College Station, TX, USA). Descriptive statistics were undertaken to describe parent and child characteristics. Sociodemographic differences in categorical survey responses were analyzed using chi-squared tests. Two-tailed *t*-tests were used for the analyses of continuous survey responses by comparing the means between sociodemographic groups. Results were considered statistically significant with *p*-values <0.05.

## 3. Results

For the larger study, approximately 60% (*n* = 463) of parents recruited consented to participate, with a final sample of 95 parents completing the questions for the current study. The participants were parents of children (52% boys) with a mean age of five years. The majority of the parents were female (82%) with a tertiary qualification (62%), an annual household income above AUD 80,000 (79%), living with a partner (89%), and have two children (61%). Sociodemographic characteristics of participants are summarized in Table 2.

Full results for questions one to six are presented in Appendix A. Table 3 presents these results as per the groupings used in the chi-square tests. The majority of children (56%, *n* = 53) consumed food from a fast-food outlet at least once a month, including 34% (*n* = 32) who did this at least once a week. In comparison, 54% (*n* = 51) of children consumed food from non-fast-food outlets at least once a month, including 18% (*n* = 17) of children who did this at least once a week. 

Of the 87 parents who reported that their child ever eats at a non-fast-food restaurant, the frequency of seeing a child menu at these outlets was often (26%, *n* = 23), sometimes (28%, *n* = 24), rarely (33%, *n* = 29), never (9%, *n* = 8) or don’t know/can’t say (3%, *n* = 3). Of the parents who had ever seen a child menu, 76 reported that their child ordered from child menus every time (7%, *n* = 5), often (29%, *n* = 22), sometimes (42%, *n* = 32) or rarely (16%, *n* = 12), while 7% (n = 5) never ordering from these menus. A similar proportion of parents felt that the portion size of items on these menus were just right (38%, *n* = 29) or too large (34%, *n* = 26), whereas 21% (n = 16) felt that the portion size was too small. The majority of parents (86%, *n* = 65) who had ever ordered from a child menu wanted to see changes to child menus at non-fast-food outlets. On average (SD), parents (*n* = 75) perceived that 45% (20%) of child menu items were healthy options but would prefer 69% (19%) of the items to be healthy options. The mean difference between perception and preference was 24% (95% CI 18% to 30%, *p* < 0.001), indicating that all participating parents would like to see a higher proportion (69%) of child menu items to be healthy as compared with what is currently offered (45%). Parents’ responses to Questions 1 to 8 were not significantly different between groups for all sociodemographic characteristics.

## 4. Discussion

The current study found that children consume food from food outlets quite regularly. Parents reported that their children consume foods from outlets classified as fast-food restaurants (at least weekly 34% and monthly 56%) and from bistros, cafes, and other non-fast-food restaurants (at least weekly 18% and monthly 54%). At non-fast food outlets, over half of parents see a child menu often or sometimes and over a third order from a child menu often or every time, with a further 42% doing this sometimes. These findings are consistent with our observations of the proportion of non-fast-food outlets in Australia offering a regular menu along with a child menu and those offering only a single menu, where patrons can share dishes. 

Although this study identified that parents’ perspectives on portion sizes were mixed, it did confirm that in the current study parental support for improving the dietary quality of child menus was unanimous. As availability is an important prerequisite of purchasing and consumption of healthy food choices, comprehensive food policies targeting greater availability of healthy child menu items are needed. Previous studies [18,19] have shown that the availability of healthier food options in restaurants can be increased by adapting current menu items into healthier alternatives; increasing the number of fruit and vegetable options, especially in main dishes and desserts; and providing low-priced healthy options. Regular training of restaurant staff regarding healthy eating has also been shown to be a successful strategy for increasing the availability of healthy options on the menu [19]. Studies have also found that changes to menus are possible without removing consumer choices or reducing revenue; affirming that such modifications have the potential to improve nutritional quality while allowing food outlets to remain competitive in the marketplace [18].

The current study also showed that parents are pragmatic in regard to the proportion of healthy child menu options they would like to see offered at non-fast-food outlets, because on average they indicated they would like to see 69% of items as ”healthy” choices. Research indicates that parents have more stringent expectations related to overall social behaviour, yet more permissive or lenient practices related to food when eating at restaurants compared to home. For example, several mothers stated they consistently limited or restricted soda and caffeinated beverages at home, but were more likely to allow their children to consume these beverages while at restaurants [20]. Studies have identified that parents want increased access to healthy foods through an increase in local healthy food outlets [21]. In Australia, programs to improve menu options in general settings (Healthier Choices Canberra) [22], health facilities (Healthy Food and Drink in NSW Health Facilities for Staff and Visitors Framework) [23] or child menus (Healthy Kids Menu, South Australian Government) [24] exist in some jurisdictions, however these have not been widely implemented across Australia [25]. It is a limitation of the current study that we only assessed parents’ perceptions of the frequency of healthy items on child menus. Limited contemporary audit data of children’s menus at non-fast-food restaurants exist in Australia [13] and this is identified as an area for future research. Furthermore, it appears that the state of the international evidence regarding promotion of healthier meals prepared out-of-home are generally of low methodological quality with many limited to assessments of acceptability [26]. The accumulating evidence supports the need to continue to develop, implement, and evaluate strategies to facilitate non-fast-food restaurants to sustainably adopt healthy food policies to improve the dietary quality of their menus. 

Care must be taken when comparing studies about “eating out” due to important differences in questionnaire items [6]. For example, in the current study we asked about how often children had meals or snacks from fast food and separately from non-fast-food outlets, regardless of the location where they consumed the food (i.e., eat in or at home via takeaway or home delivery). In a large UK study of data collected between 2008 and 2012 from a sample including 1963 children (1.5 to 18 years old), the questions focused on the eating location, for example, “On average, how often do you/does child eat meals out in a restaurant or cafe?” and “On average, how often do you/ does child eat take-away meals at home?” [27]. In that study, meals were defined as more than a beverage or packet of chips, with 19% reporting eating meals out at least weekly and 21% having takeaway meals at home at least weekly [27]. Unlike the current study which has a smaller sample size and did not identify any sociodemographic patterns in children’s eating out behaviors, this UK-based study found that children from less affluent households were more likely to eat take-away meals at home once per week or more [27]. A cross-sectional study of adults in the UK also reported that lower socioeconomic status, measured by lower educational attainment and household income, was associated with more frequent consumption of take-away meals, however, higher socioeconomic status was associated with eating out more frequently [28]. 

A strength of the current study is that parents were asked about the frequency of their child’s intake of food from fast-food outlets first and then the remaining questions asked parents to consider only bistros, cafes, and other non-fast-food restaurants. We believe this technique has allowed for a clear definition of non-fast-food outlets for parents. The consumption of foods from non-fast-food restaurants in various locations, for example, eat-in, takeaway, or home delivered foods, was also included to more accurately reflect contemporary dining options. Hence, data reported in this study are distinct within the limited evidence base for parent views on child menus in the non-fast-food sector. There are several limitations of this study with regards to generalizability of findings and potential sources of bias. The Australian Bureau of Statistics data indicate that 35% to 41% of women aged 25 to 44 years have a tertiary qualification [29], 56% of Australian households have a gross income >$80,000 [30], and 22% of families have one parent [31]. The sociodemographic characteristics of participants in this small convenience sample of families living in NSW were predominately tertiary educated (62%), mothers (82%), from dual-parent households (89%), with a household annual income >$80,000 (79%). Hence, research on a more representative population sample is required before results can be generalized to the population nationally. The results likely present a best-case scenario, and further evaluation of the survey questions regarding parents’ views of child menus in samples with a greater representation of parents who have a lower educational attainment, a lower household income, and in fathers is warranted. Data from a South Australian study found that 57% of consumers perceived healthy child meal options as more expensive than less healthy options, but over a third thought price was the same [13]. In that study [13], two-thirds of consumers indicated they would be willing to pay up to 10% more for healthier options. Nevertheless, the collection of objective and representative data on the dietary quality of child menus at non-fast-food outlets in Australia is limited and could be addressed to facilitate action in this domain. 

In conclusion, in this sample, food ordered from child menus in the non-fast-food sector appears to make a relatively small contribution to the diets of the majority of children. However, the importance of this contribution is unknown, such as the impact children’s observations of restaurant meals has on shaping food preferences their perceptions about food norms. Nonetheless, parents’ views support the implementation of initiatives to increase the availability of healthy options on child menus at non-fast-food outlets. We recommend that a comprehensive study of child menus in the non-fast-food sector should be undertaken to provide the evidence needed to secure investment in policies and interventions to improve the diet quality of child menus. Providing children and their families with more options to make healthy food choices habitual, when consuming food prepared outside the home, could help to provide greater consistency with efforts by early childhood and school settings to improve nutrition. 

## Figures and Tables

**Table 1 children-06-00123-t001:** Survey questions and response options.

Questions	Response Options
1. How often does your child/ren have meals or snacks such as burgers, pizza, chicken, or chips from places like McDonalds, Hungry Jacks, Pizza Hut, KFC, Red Rooster or local takeaway food places (include eat-in, takeaway or home delivered foods)?	Never, less than once a month, less than once a week, one to two times per week, three to four times per week, five to six times per week, or every day
2. How often does your child/ren have meals or snacks from a bistro, café or non-fast-food restaurant (include eat-in, takeaway or home delivered foods)?	As above in Question 1
3. When you have a meal or snack, with your child/ren, from a bistro, café or non-fast-food restaurant, how often do you see that a “childrens’ menu” or a “kids’ menu” is available?	Every time, often, sometimes, rarely, or never
4. When ordering a meal or snack for your child/ren (6 months to 12 years of age) from a bistro, café or non-fast-food restaurant, have you or your child/ren ever ordered from a “childrens’ menu” or a "kids’ menu"?	As above in Question 3
5. Are there any changes that you would like to see made to “childrens’ menus” or "kids’ menus" at bistros, cafés or non-fast-food restaurants?	Yes, no, or I don’t know/can’t say
6. At a typical bistro, café or non-fast food restaurant, how would you describe the overall portion size of options on “childrens’ menus” or “kids’ menus”?	Too large, too small, just right, or I don’t know
7. At a typical bistro, café or non-fast-food restaurant, what percentage of options on “childrens’ menus” or “kids’ menus” do you think are healthy options? (Scale 1–100)	Sliding scale: 0% to 100% of the menu
8. At bistros, cafés or non-fast-food restaurants, what percentage of options on “childrens’ menus” or “kids’ menus” would you like to be healthy options? (Scale 1–100)	As above in Question 7

**Table 2 children-06-00123-t002:** Sociodemographic characteristics of participants.

Characteristics (*n* = 95)	*n* (%) or Mean ± SD
Child age (years), mean ± SD	5 ± 1
Child sex ^a^	
Girl	44 (46)
Boy	49 (52)
Parent’s sex, *n* (%)	
Female	78 (82)
Male	17 (18)
Parent’s highest education level, *n* (%)	
Higher School Certificate	8 (8)
TAFE Certificate or Diploma	28 (30)
University or other tertiary institutes	59 (62)
Household annual income in AUD ^a^, *n* (%)	
$20,000–$40,000	3 (3)
$40,001–$60,000	4 (4)
$60,001–$80,000	9 (9)
More than $80,000	75 (79)
Family context, *n* (%)	
I have a shared care arrangement	3 (3)
I live on my own raising children	4 (4)
I live with another person/people raising children	3 (3)
I live with my partner raising children	85 (89)
Number of children in the family, *n* (%)	
1	12 (13)
2	58 (61)
3	20 (21)
4–6	5 (5)

*n*: Number; TAFE: Technical and Further Education; AUD: Australian dollars. ^a^ Percentages may not sum up to 100% due to missing data or rounding.

**Table 3 children-06-00123-t003:** Grouped survey responses, *n* (%), stratified by dichotomized sociodemographic characteristics.

	Total Sample	Child Sex		Parent’s Sex		Parental Education		Household Annual Income		Number of Children	
		Girl	Boy	Female	Male	High school/TAFE/Diploma	University/tertiary institute	≤$80,000	>$80,000	1 to 2	≥3
Q1											
<Once a month	42 (44)	20 (45)	21 (43)	35 (45)	7 (41)	15 (42)	27 (46)	5 (31)	35 (47)	33 (47)	9 (36)
≥Once a month	53 (56)	24 (55)	28 (57)	43 (55)	10 (59)	21 (58)	32 (54)	11 (69)	40 (53)	37 (53)	16 (64)
Total	95 (100)	44 (100)	49 (100)	78 (100)	17 (100)	36 (100)	59 (100)	16 (100)	75 (100)	70 (100)	25 (100)
Q2											
<Once a month	44 (46)	24 (55)	18 (37)	35 (45)	9 (53)	18 (50)	26 (44)	5 (31)	38 (51)	35 (50)	9 (36)
≥Once a month	51 (54)	20 (45)	31 (63)	43 (55)	8 (47)	18 (50)	33 (56)	11 (69)	37 (49)	35 (50)	16 (64)
Total	95 (100)	44 (100)	49 (100)	78 (100)	17 (100)	36 (100)	59 (100)	16 (100)	75 (100)	70 (100)	25 (100)
Q3											
Sometimes/rarely/never	64 (74)	28 (74)	34 (72)	51 (72)	13 (81)	21 (68)	43 (77)	10 (77)	51 (73)	48 (75)	16 (70)
Everytime/often	23 (26)	10 (26)	13 (28)	20 (28)	3 (19)	10 (32)	13 (23)	3 (23)	19 (27)	16 (25)	7 (30)
Total	87 (100)	38 (100)	47 (100)	71 (100)	16 (100)	31 (100)	56 (100)	13 (100)	70 (100)	64 (100)	23 (100)
Q4											
Sometimes/rarely/never	49 (64)	22 (67)	27 (66)	37 (60)	12 (86)	16 (57)	33 (69)	8 (67)	39 (64)	36 (63)	13 (68)
Everytime/often	27 (36)	11 (33)	14 (34)	25 (40)	2 (14)	12 (43)	15 (31)	4 (33)	22 (36)	21 (37)	6 (32)
Total	76 (100)	33 (100)	41 (100)	62 (100)	14 (100)	28 (100)	48 (100)	12 (100)	61 (100)	57 (100)	19 (100)
Q5											
Yes	65 (86)	28 (85)	35 (85)	54 (87)	11 (79)	26 (93)	39 (81)	12 (100)	51 (84)	49 (86)	16 (84)
No	8 (11)	4 (12)	4 (10)	5 (8)	3 (21)	1 (4)	7 (15)	0 (0)	7 (11)	5 (9)	3 (16)
I don’t know/can’t say	3 (4)	1 (3)	2 (5)	3 (5)	0 (0)	1 (4)	2 (4)	0 (0)	3 (5)	3 (5)	0 (0)
Total	76 (100)	33 (100)	41 (100)	62 (100)	14 (100)	28 (100)	48 (100)	12 (100)	61 (100)	57 (100)	19 (100)
Q6											
Too large/small	42 (55)	14 (42)	26 (63)	34 (55)	4 (29)	15 (54)	27 (56)	6 (50)	35 (57)	31 (54)	11 (58)
Just right	29 (38)	16 (48)	13 (32)	25 (40)	4 (29)	12 (43)	17 (35)	5 (42)	22 (36)	22 (39)	7 (37)
I don’t know	5 (7)	3 (9)	2 (5)	3 (5)	2 (14)	1 (4)	4 (8)	1 (8)	4 (7)	4 (7)	1 (5)
Total	76 (100)	33 (100)	41 (100)	62 (100)	14 (100)	28 (100)	48 (100)	12 (100)	61 (100)	57 (100)	19 (100)

Q: Question; TAFE: Technical and Further Education. Results are presented in *n* (%), the total may not sum up to 100% due to missing responses in incomplete surveys or rounding. The survey questions were: Q1. How often does your child/ren have meals or snacks such as burgers, pizza, chicken, or chips from places like McDonalds, Hungry Jacks, Pizza Hut, KFC, Red Rooster or local takeaway food places? (include eat-in, takeaway or home delivered foods). Q2. How often does your child/ren have meals or snacks from a bistro, café or non-fast-food restaurant (include eat-in, takeaway or home delivered foods). Q3. When you have a meal or snack, with your child/ren, from a bistro, café or non-fast-food restaurant, how often do you see that a “childrens’ menu” or a “kids’ menu” is available? Q4. When ordering a meal or snack for your child/ren (6 months to 12 years of age) from a bistro, café or non-fast-food restaurant, have you or your child/ren ever ordered from a “childrens’ menu” or a "kids’ menu"? Q5. Are there any changes that you would like to see made to “childrens’ menus” or "kids’ menus" at bistros, cafés or non-fast-food restaurants? Q6. At a typical bistro, café or non-fast-food restaurant, how would you describe the overall portion size of options on “childrens’ menus” or “kids’ menus”?

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
