# Peer review of "Children’s Intake of Food from Non-Fast-Food Outlets and Child-Specific Menus: A Survey of Parents"

_children, 2019, doi:10.3390/children6110123_

Round 1

Reviewer 1 Report

The manuscript by Li Kheng Chai et al reports parental responses to a questionnaire on child menu awareness, perception, and selection in non-fast-food restaurants in a convenience sample from NSW, Australia.

Major comments:

Considering that the sample is substantially biased in favour of high(er)-SES parents and may also suffer from potential social desirability bias, the findings are probably best regarded as very preliminary. An objective assessment of the healthfulness of the kids' menus in question would be necessary to understand how much room for improvement there is. In this context, the "unanimous parental support for improving the dietary quality of child menus" is a clear overstatement of the results. The reporting of majorities in the paragraph starting l. 119 further compounds the impression of over-interpretation of the data collected. Why did the authors not follow their own grouping of responses as provided in Table 2, where for Q3 and Q4 the more appropriate grouping into "sometimes/rarely/never" and "everytime/often" was chosen?

The authors might wish to consider complementing the findings with a broader roll-out to a representative sample and using objective assessments of menu healthfulness. This would substantially strengthen the scientific and policy value of the research. You could also consider studying to what extent parents' calls for healthier restaurant menu offers are in line with what is being practiced in their homes. It is easy to make the most outrageous claims for healthy foods, 100% organic, etc. when the responsibility for providing such a service lies elsewhere. And if such food options cost more, would parents really be willing to pay the higher price?

Minor comments:

l. 22: 78% of 100% or of the 87% that reported seeing a child menu? Logically it would have to be the latter, but the phrasing could be improved to remove all doubt. At the same time, reporting a percentage of a percentage risks losing the reader. I would suggest to at least include actual numbers of respondents.

l. 38: Unclear to me what "greater proportion" you mean.

ll. 51-52: Suggest to revise phrase to "For Australia, there is limited data in the peer-reviewed literature." or similar.

l. 55: What percentage could have ordered a healthier kids' meal, i.e. in how many instances were healthier options available?

l. 83ff: Suggest to clearly identify/list the questions and their numbers in the text.

l. 95: parents' perception of the healthfulness of menu options seems a very bias-prone indicator of actual healthfulness.

l. 101: Please indicate if the t-test was one- or two-tailed, homo- or heteroscedastic.

Table 1: Child age does not fit the column description of "N (%)". The total number of children (n=93) in the sample seems to contradict the statement that the majority of parents (61%) has two children. Overall, the sample is heavily biased with 79% belonging to the highest income quartile.

l. 173: Suggest to delete "Adam et al's".

Author Response

Response to reviewers’ comments

We would like to thank the reviewers for their thoughtful comments on the manuscript. We have revised the manuscript as recommended and provided a point-by-point response to the reviewers’ comments in this document.

Reviewer 1

Major comments:

Point 1: Considering that the sample is substantially biased in favour of high(er)-SES parents and may also suffer from potential social desirability bias, the findings are probably best regarded as very preliminary. An objective assessment of the healthfulness of the kids' menus in question would be necessary to understand how much room for improvement there is.

Response 1: Thank you for highlighting that greater attention to further addressing the study limitations is needed. We have edited the discussion as follows:

“There are several limitations of this study with regards to generalizability of findings and potential sources of bias. Australian Bureau of Statistics data indicate that 35-41% of women aged 25-44 years have a tertiary qualification, 56% of Australian households have a gross income >$80K and 22% of families had one parent. Therefore, the sociodemographic characteristics of participants in this small convenience sample of families living in NSW were predominately tertiary educated (62%), mothers (82%), from dual-parent households (89%), with a household annual income >$80k (79%). Hence, further research on a more representative population sample is required before results can be generalized to the population nationally. The results likely present a best case scenario and further evaluation of the survey questions regarding parent’s views of child menus in samples with a greater representation of parents who have a lower educational attainment, a lower household incomes and in fathers is warranted. Other limitations include that social desirability bias may have influenced parent responses regarding improvements to the child menus and that whether cost or value for money were factors influencing decisions made about ordering from child menus were not addressed.”

Consistent with our observations of child menus locally and in other metropolitan and regional areas, we have identified one contemporary Australian report regarding items on child menus. This report has now been referenced and identified that child menus across 35 clubs most commonly offered chicken nuggets and chicken schnitzel as meals, with chips being the most prevalent and usually the default side dish, while ice cream was the most common dessert option, with no healthy desserts available, and soft-drink the most common beverage option. We have added to the discussion as follows:-

“It is a limitation of the current study that we only assessed parent’s perceptions of the frequency of healthy items on child menus. Limited contemporary audit data of children’s menus at non-fast food restaurants exist in Australia (ref: Benchmark research) and this is identified as an area for future research.”

Point 2: In this context, the "unanimous parental support for improving the dietary quality of child menus" is a clear overstatement of the results. The reporting of majorities in the paragraph starting l. 119 further compounds the impression of over-interpretation of the data collected. Why did the authors not follow their own grouping of responses as provided in Table 2, where for Q3 and Q4 the more appropriate grouping into "sometimes/rarely/never" and "everytime/often" was chosen?

Response 2: We have added data to the results and revised the statement to avoid over-interpretation now reads:

“…..in the current study parental support for improving the dietary quality of child menus was unanimous”.

We have now removed the reporting of ‘majorities’ in the paragraph starting at line 119 and reported the % in text for each response category separately, which now reads:

“Of the 87 parents who reported that their child ever eats at a non-fast food restaurant, the frequency of seeing a child menu at these outlets was often (26%; n=23), sometimes (28%; n=24), rarely (33%; n=29), never (9%; n=8) or don’t know/can’t say (3%; n=3).”

Point 3: The authors might wish to consider complementing the findings with a broader roll-out to a representative sample and using objective assessments of menu healthfulness. This would substantially strengthen the scientific and policy value of the research. You could also consider studying to what extent parents' calls for healthier restaurant menu offers are in line with what is being practiced in their homes. It is easy to make the most outrageous claims for healthy foods, 100% organic, etc. when the responsibility for providing such a service lies elsewhere. And if such food options cost more, would parents really be willing to pay the higher price?

Response 3: While we agree that it is an excellent suggestion to complement the survey findings with a representative sample with objective assessments of menu healthfulness. We feel that is beyond the scope of the current study and could be addressed by future research. However, we strongly agree with the essence of the reviewer’s point and have added the following statement to the discussion:

“However, the collection of objective and representative data on the dietary quality of child menus at non-fast food outlets in Australia is lacking and could be addressed to facilitate action in this domain.”

Minor comments:

Point 4: l. 22: 78% of 100% or of the 87% that reported seeing a child menu? Logically it would have to be the latter, but the phrasing could be improved to remove all doubt. At the same time, reporting a percentage of a percentage risks losing the reader. I would suggest to at least include actual numbers of respondents.

Response 4: We have included actual numbers of respondents as suggested. The manuscript has been revised to address comment in Points 2 and 4, and now reads: “Of the 87 parents who reported that their child ever eats at a non-fast food restaurant, the frequency of seeing a child menu at these outlets was often (26%; n=23), sometimes (28%; n=24), rarely (33%; n=29), never (9%; n=8) or don’t know/can’t say (3%; n=3).”

Point 5: l. 38: Unclear to me what "greater proportion" you mean.

Response 5: The manuscript has been revised to improve clarity and this now reads:

“It was identified that food consumed in these settings contribute a greater proportion of daily total energy intake from energy-dense, nutrient-poor, non-core foods compared to nutrient-dense, core foods, with the opposite reported in home and school settings [5].”

Point 6: l. 51-52: Suggest to revise phrase to "For Australia, there is limited data in the peer-reviewed literature." or similar.

Response 6: The phrase has been revised to: "In Australia, there is currently limited data in the peer-reviewed literature."

Point 7: l. 55: What percentage could have ordered a healthier kids' meal, i.e. in how many instances were healthier options available?

Response 7: The manuscript has been revised to improve clarity and now reads:

“Results indicated that almost two-thirds of child meals were ordered from child menus, with 8% ordering healthier kids' meals, despite healthier options being available in all participating food outlets.”

Point 8: l. 83: Suggest to clearly identify/list the questions and their numbers in the text.

Response 8: A new table (Table 1) has been added to list the relevant survey questions to improve readership.

Point 9: l. 95: parents' perception of the healthfulness of menu options seems a very bias-prone indicator of actual healthfulness.

Response 9: The initial aim of the current study was to learn about consumer (i.e. parent) views on child menus in the non-fast food sector. We acknowledge as per our response to point 1 that it is a limitation that we did not also collect objective data on child menus across the study region. This task was beyond the resources available for this study, but is a future research aim and will be informed by the current results. 

Point 10: l. 101: Please indicate if the t-test was one- or two-tailed, homo- or heteroscedastic.

Response 10: Additional information has been added to the manuscript to improve clarity, and now reads: “Two-tailed t-tests were used to analyse continuous survey response by comparing the means between socio-demographic groups.”

Point 11: Table 1: Child age does not fit the column description of "N (%)".

Response 11: The column description has been revised to “N (%) or mean ±SD”.

Point 12: Table 1: The total number of children (n=93) in the sample seems to contradict the statement that the majority of parents (61%) has two children.

Response 12: The number of children in the family was collected as part of family demographic. However, the survey included responses from one index child (who met the age criteria) in each family who participated in the survey. Additional information has been added to the manuscript to improve clarity, and this now reads:

“A sub-sample of parents were randomly selected to receive these questions and were asked to respond with regard to their index child attending the childcare service.”

Point 13: Table 1: Overall, the sample is heavily biased with 79% belonging to the highest income quartile.

Response 13: We have now addressed this comment in our response to Reviewer 1 Point 1.

Point 14: l. 173: Suggest to delete "Adam et al's".

Response 14: "Adam et al's" has been deleted as suggested.

Reviewer 2

Point 15: One of my first impressions was that this was a small sample size, but later in the manuscript, the authors acknowledged this and also stated that this was a limitation of the study.  Thus, the conclusions may not be very generalizable.

Response 15:  We have addressed this comment in our response to Reviewer 1 Point 1. The revised discussion now reads:

“Hence, research on a more representative population sample is required before results can be generalized to the population nationally. The results likely present a best case scenario and further evaluation of the survey questions regarding parent’s views of child menus in population groups with samples with a greater representation of parents who have a lower educational attainment, a lower household income of parents and in fathers is warranted.”

Point 16: On page 2, line 52, "in the" is repeated

Response 16: The repeated phrase "in the" has been removed.

Point 17: I would like to know if there was an incentive for participation in the study.

Response 17: There was no incentive for participation in the study, and the information has now been added in the manuscript, which now reads: “No incentive was provided to participants in the study.”

Point 18: In publications in the US, they often include the year and version of the data analysis program that was used.

Response 18: The year and version of the data analysis program has been added to the manuscript, and now reads: “All data manipulation and statistical analyses was undertaken using STATA version 12 (StataCorp LLC).”

Point 19: In Table 1, consider providing a legend which defines "AUD."

Response 19: A legend has been added to define “AUD”, which is Australian dollars.

Point 20: In Table 1, the third line under "Family Context" seems to be incomplete.

Response 20: The phrase has been revised, and now reads: “I live with another person/people raising children”.

Point 21: It wasn't clear to me if the participants were asked if price/cost was a factor in deciding whether to order from the child's menu.  That may have played a larger role than the nutritional value of the menu. Often times, menu prices on the children's menus are lower. Other times, buying the adult size portion and splitting it between children may be more economical.

Response 21: Participants were not asked about price or cost factors in this study. However, we suggest that future research should explore this factor. Additional information has been added to the manuscript to improve the clarity, and it now reads:

“Other limitations are that social desirability bias may have influenced parent’s responses regarding improvements to the child menus. Further, we did not address whether cost or value for money were factors in decisions made about ordering from child menus. Data from a South Australian study found that 57% of consumers perceived healthy child meal options as more expensive than less healthy options, but over a third thought price was the same. In that study, two-thirds of consumers indicated they would be willing to pay up to 10% more for healthier options.”

Point 22: Overall, it is generally well written article, but the results did not seem to add a lot to the current literature on this issue.  

Response 22: The manuscript contributes to a novel and important area of public health which is currently understudied, by exploring young children’s frequency of consuming foods at non-fast food outlets, their parent’s observations of child menus at these outlets and their purchasing behaviours/future preferences regarding child menus.

This study provides important consumer perspectives to inform future research, including future interventions to improve child menus in the non-fast food sector. We have also extended the strengths section of the manuscript:

“A strength of the current study is that parents were asked about the frequency of their child’s intake of food from fast-food outlets first and then the remaining questions asked parents to consider only bistros, cafes, and other non-fast food restaurants. We believe this technique has allowed a clear definition of what parents meant by non-fast food outlets. The consumption of foods from non-fast food restaurants in various locations e.g. include eat-in, takeaway or home delivered foods, was also included to more accurately reflect contemporary dining options.  Hence, data reported in the current study are distinct within the limited evidence base on parent views on child menus in the non-fast food sector.”

Reviewer 2 Report

One of my first impressions was that this was a small sample size, but later in the manuscript, the authors acknowledged this and also stated that this was a limitation of the study.  Thus, the conclusions may not be very generalizable. On page 2, line 52, "in the" is repeated I would like to know if there was an incentive for participation in the study. In publications in the US, they often include the year and version of the data analysis program that was used. In Table 1, consider providing a legend which defines "AUD." In Table 1, the third line under "Family Context" seems to be incomplete. It wasn't clear to me if the participants were asked if price/cost was a factor in deciding whether to order from the child's menu.  That may have played a larger role than the nutritional value of the menu. Often times, menu prices on the children's menus are lower.  Other times, buying the adult size portion and splitting it between children may be more economical. Overall, it is generally well written article, but the results did not seem to add a lot to the current literature on this issue.  

Author Response

Response to reviewers’ comments

We would like to thank the reviewers for their thoughtful comments on the manuscript. We have revised the manuscript as recommended and provided a point-by-point response to the reviewers’ comments in this document.

Reviewer 2

Point 15: One of my first impressions was that this was a small sample size, but later in the manuscript, the authors acknowledged this and also stated that this was a limitation of the study.  Thus, the conclusions may not be very generalizable.

Response 15:  We have addressed this comment in our response to Reviewer 1 Point 1. The revised discussion now reads:

“Hence, research on a more representative population sample is required before results can be generalized to the population nationally. The results likely present a best case scenario and further evaluation of the survey questions regarding parent’s views of child menus in population groups with samples with a greater representation of parents who have a lower educational attainment, a lower household income of parents and in fathers is warranted.”

Point 16: On page 2, line 52, "in the" is repeated

Response 16: The repeated phrase "in the" has been removed.

Point 17: I would like to know if there was an incentive for participation in the study.

Response 17: There was no incentive for participation in the study, and the information has now been added in the manuscript, which now reads: “No incentive was provided to participants in the study.”

Point 18: In publications in the US, they often include the year and version of the data analysis program that was used.

Response 18: The year and version of the data analysis program has been added to the manuscript, and now reads: “All data manipulation and statistical analyses was undertaken using STATA version 12 (StataCorp LLC).”

Point 19: In Table 1, consider providing a legend which defines "AUD."

Response 19: A legend has been added to define “AUD”, which is Australian dollars.

Point 20: In Table 1, the third line under "Family Context" seems to be incomplete.

Response 20: The phrase has been revised, and now reads: “I live with another person/people raising children”.

Point 21: It wasn't clear to me if the participants were asked if price/cost was a factor in deciding whether to order from the child's menu.  That may have played a larger role than the nutritional value of the menu. Often times, menu prices on the children's menus are lower. Other times, buying the adult size portion and splitting it between children may be more economical.

Response 21: Participants were not asked about price or cost factors in this study. However, we suggest that future research should explore this factor. Additional information has been added to the manuscript to improve the clarity, and it now reads:

“Other limitations are that social desirability bias may have influenced parent’s responses regarding improvements to the child menus. Further, we did not address whether cost or value for money were factors in decisions made about ordering from child menus. Data from a South Australian study found that 57% of consumers perceived healthy child meal options as more expensive than less healthy options, but over a third thought price was the same. In that study, two-thirds of consumers indicated they would be willing to pay up to 10% more for healthier options.”

Point 22: Overall, it is generally well written article, but the results did not seem to add a lot to the current literature on this issue.  

Response 22: The manuscript contributes to a novel and important area of public health which is currently understudied, by exploring young children’s frequency of consuming foods at non-fast food outlets, their parent’s observations of child menus at these outlets and their purchasing behaviours/future preferences regarding child menus.

This study provides important consumer perspectives to inform future research, including future interventions to improve child menus in the non-fast food sector. We have also extended the strengths section of the manuscript:

“A strength of the current study is that parents were asked about the frequency of their child’s intake of food from fast-food outlets first and then the remaining questions asked parents to consider only bistros, cafes, and other non-fast food restaurants. We believe this technique has allowed a clear definition of what parents meant by non-fast food outlets. The consumption of foods from non-fast food restaurants in various locations e.g. include eat-in, takeaway or home delivered foods, was also included to more accurately reflect contemporary dining options.  Hence, data reported in the current study are distinct within the limited evidence base on parent views on child menus in the non-fast food sector.”
